# Effects of Dam Building on Niche Differentiation of Comammox *Nitrospira* in the Main Stream of the Three Gorges Reservoir Area

Shuang Liu [1,2,3,†], Jiahui Zhang [1,2,3,†], Yuchun Wang [1,2], Mingming Hu [1,2,*], Yufei Bao [1,2], Shanze Li [1,2], Jie Wen [1,2] and Jianwei Zhao [3,*]

[1] State Key Laboratory of Simulation and Regulation of Water Cycle in River Basin, Beijing 100038, China
[2] Department of Water Ecology and Environment, China Institute of Water Resources and Hydropower Research, Beijing 100038, China
[3] State Environmental Protection Key Laboratory of Soil Health and Green Remediation, College of Resources and Environment, Huazhong Agricultural University, Wuhan 430070, China
* Correspondence: hmmkeke@163.com (M.H.); jwzhao2@163.com (J.Z.); Tel.: +86-10-6878176 (M.H.); +86-27-85669068 (J.Z.)
† These authors contributed equally to this work.

**Abstract:** Complete ammonia oxidizers (comammox) can completely oxidize ammonia to nitrate, and the various habitats of comammox *Nitrospira* are an important guarantee for their survival. The construction of the Three Gorges Dam, China, made it easier for nitrogen to stay in the reservoir area, which may have caused the niche differentiation of comammox, thereby affecting the natural transformation process of nitrogen in the reservoir area. This study investigated comammox in river sediments in the Three Gorges Reservoir area. Comammox clade A and comammox clade B were detected in all samples, and comammox clade A was dominant. The number of dominant OTUs (Operational Taxonomic Unit) in comammox clade A.1 accounted for 18.69% of the total number of OTUs, followed by comammox clade A.2 (18.58%) and clade B (14.30%). The indicated abundance of comammox *Nitrospira* clade A and clade B *amoA* genes in the main stream of the Three Gorges Reservoir increased along the length of the river and reached the maximum in the middle part of the reservoir area. The highest abundance of ammonia-oxidizing archaea (AOA) *amoA* genes appeared in the upper stream section of the reservoir area. Comammox *Nitrospira* clade A exhibited the highest abundance ($3.00 \times 10^4 \pm 8782.37$ copies/g), followed by comammox *Nitrospira* clade B ($1.83 \times 10^3 \pm 1019.82$ copies/g), ammonia-oxidizing bacteria (AOB) ($1.28 \times 10^3 \pm 574.69$ copies/g), and AOA ($1.73 \times 10^2 \pm 48.05$ copies/g). The abundances of both comammox clades A and B were positively correlated with sediment pH, indicating that pH is an important environmental factor affecting the growth of comammox bacteria. Additionally, the relative abundances of both comammox clade A.2 and clade B were significantly correlated with overlying water dissolved oxygen (DO) in the reservoir area. This study thus indicated that there exists a niche differentiation of comammox *Nitrospira* in the main stream of the Three Gorges Reservoir area. The potential changes in the ammoxidation process and the environmental effects caused by this niche differentiation need further attention.

**Keywords:** comammox; niche differentiation; nitrification; AOA; AOB

## 1. Introduction

Nitrification is a key link in the transformation of nitrogen in water [1]. Nitrification generally includes two steps, namely, ammonia oxidation and nitrite oxidation. As the rate-limiting step, ammonia oxidation is completed by ammonia-oxidizing bacteria (AOB) and ammonia-oxidizing archaea (AOA), with nitrite oxidation completed by nitrite-oxidizing bacteria (NOB) [2,3]. Before 2015, it was generally agreed that traditional nitrification was

completed in two steps. However, in 2015, Daims et al. and van Kessel et al. found that there were microorganisms that could directly oxidize ammonia ($NH_3$) to nitrate, which were called complete ammonia oxidizer (comammox) bacteria [4,5]. These discoveries changed the traditional two-step nitrification theory. All the comammox bacteria found so far affiliated to *Nitrospira* Lineage II and they can be divided into clade A and clade B. Both clade A and clade B can realize ammonia oxidation. However, comammox clade A is usually dominant in aquatic environments [6], while clade B has advantages in forest, cropland and other environment [7,8].

There is competition for ammonia between traditional ammonia-oxidizing microorganisms and comammox *Nitrospira*, which may lead to their niche differentiation. Niche differentiation has been shown in a variety of ecosystems, including rivers and soils [7,9]. Farmland soil studies have shown that there are niche differences between AOA and AOB, and there are differences between comammox bacteria and AOA and AOB in niches with different pH. The nitrification activities of AOA and AOB depend on pH, and comammox can also be active in acidic soil [10]. One study of soil aggregates revealed that comammox *Nitrospira* can coexist with both AOA and AOB, and their niche distribution is more similar to that of AOA [11]. Another study found that clade A and clade B of comammox *Nitrospira* exhibit differences in ammonia affinity and ecological conditions [12], and thus, clades A and B may also be differentiated. A recent study has shown that clade A and clade B could differentiate in farmland soil and tidal flats in the Yangtze River estuary, furthermore, clade A could differentiate into three sub-clades: A.1, A.2, and A.3 [7,13–15]. When the water environment changes spatially, the survival status of each clade of comammox bacteria may change, resulting in the phenomenon of differentiation. The results of RNA transcription indicated that comammox bacteria had a high contribution to the nitrification of sewage treatment plants [16]. Comammox bacteria were also found to play an important role in aquatic ecosystems and terrestrial soils, and their contribution was higher than that of AOA and AOB in some areas [17]. Furthermore, pH may have an effect on each clade of comammox bacteria [18,19], but it remains unclear how the comammox *Nitrospira* clades compete with each other and coexist.

Cascade dam construction alters the transport of nutrients in the river, making it easier for sediment to settle down in the reservoir area and thus resulting in the accumulation of nitrogen and phosphorus in front of the reservoir [20–22]. Previous work has shown that dam construction and impoundment will lead to the accumulation of nitrogen and phosphorus in the river and the enrichment of nutrients in the river, thus changing the living environment of microorganisms [23]. The Three Gorges Reservoir (TGR) is the largest reservoir in the world and is located in the upper reaches of the Yangtze River in China, with a total water storage capacity of 39.3 billion $m^3$. The total length of the Three Gorges Reservoir is 663 km, and the water area reaches 1084 $km^2$ [24]. Every year, the Three Gorges Reservoir undergoes a periodic storage and discharge process [25]. Water is stored in the dry season and drained in the rainy season. The upstream sediments are easily deposited and bring a large amount of nutrients [26], and so the main stream in the reservoir area changes the nutrient concentration gradient along the river.

In our previous study, we found that there was a significant positive correlation between comammox *Nitrospira* clade A *amoA* gene abundance and flow distance [14]. The construction of the Three Gorges Dam has resulted in spatial changes in the main stream sediments, which may contribute to differentiation in the comammox. However, there is still a lack of understanding of whether comammox bacteria show differentiation in the main stream of the Three Gorges Reservoir area and what the main influencing factors are. This study proposed the hypothesis that the construction of Three Gorges Dam might have caused niche differentiation of the comammox *Nitrospira* in the reservoir area. To test this hypothesis, we investigated the abundance, diversity, and distribution of comammox *Nitrospira* in the main stream sediments of the Three Gorges Reservoir area of the Yangtze River as well as their competition relationship with traditional ammonia-oxidizing microorganisms.

## 2. Materials and Methods

### 2.1. Sample Collection

The study area was the main stream of the Three Gorges Reservoir area (30°56′57.24″ ~31°5′47.89″ N, 108°39′27.34″~108°42′52.91″ E), China. It has a typical subtropical monsoon climate with warm, less rainy winters and warm rainy summers. In this study, 9 sampling points (M1–9) were selected from the main stream of the Three Gorges Reservoir region, covering the area from the beginning to the end of the reservoir (Figure 1). There are nine administrative regions from the beginning to the end of the reservoir, and one or two sampling points were arranged in each region depending on the area of the region (M1–9). In December 2019, the temperature, pH, and DO of the overlying water were measured with a YSI water quality meter (EXO$_2$, Ohio City, OH, USA). Three samples of 10 cm surface sediments were collected from each sampling point, packed in sterile ziplock bags, stored on ice, and transported to the laboratory. Each sediment sample was divided into two parts, with one part used for the determination of physicochemical properties and the other part stored at −80 °C for the determination of microbial community composition.

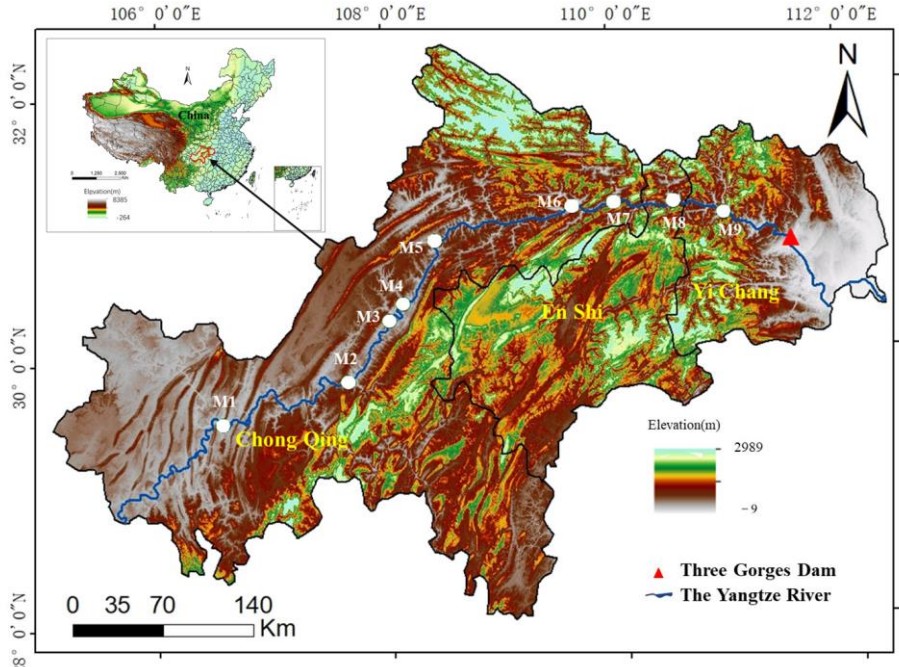

**Figure 1.** Location of sampling sites in the Three Gorges Reservoir area. The small map in the upper left corner is a map of China, and the large map is a map of the Three Gorges Reservoir. The basic geographic information was obtained from the National Geomatics Center of China (NGCC) and generated using ArcGIS 10.7 (http://www.esri.com/, accessed on 25 January 2022). M1–9 represents 9 sampling points in the main stream of the reservoir.

### 2.2. Determination of Soil Physicochemical Properties

The fresh sediment was dried at 105 °C to a constant weight to determine the moisture content. The sediment pH was determined using a pH meter (Mettler Toledo S400-B, Mettler Toledo, Zurich, Switzerland) at a sediment/water suspension ratio of 1:2.5 (*w/v*). Ten grams of air-dried sediment was leached with 50 mL 2 mol/L KCl, and then, the ammonia nitrogen (NH$_4^+$-N), nitrate nitrogen (NO$_3^-$-N), and nitrite nitrogen (NO$_2^-$-N) levels of the filtrate were determined using a flow analyzer (AutoAnalyzer 3, Seal Analytical GmbH, Norderstedt, Germany). The total carbon (TC) and total nitrogen (TN) in the sediments were determined using an elemental analyzer (Elementar Vario PYRO cube, Elementar Analysensysteme GmbH, Hanau, Germany).

### 2.3. DNA Extraction and PCR Amplification

Microbial DNA was extracted from a 0.4 g sediment sample using the Fast DNA Spin kit for soil (MPBIO, Santa Ana, CA, USA) according to the kit manufacturer's instructions. The quality of the DNA extract was inspected using a super differential spectrophotometer (NanoPhotometer-N60, IMPLEN, Munich, Germany).

The specific primers Com-*amoA*_1_R (CGAGATCATGGTGCTGTGAC) [27] and pmoA-189b F (GGNGACTGGGACTTYTGG) were used to amplify the sediment comammox *amoA* gene [27]. The total PCR amplification system (25 μL) (Applied Biosystems, Waltham, MA, USA) contained 2 μL of template DNA, 5 μL of 5 × reaction buffer, 2 μL of dNTP (2.5 mM, TransGen, Bejing, China), 1 μL of each primer (10 μM), 5 μL of 5 × GC buffer, 0.25 μL of Q5® High-Fidelity DNA Polymerase (New England Biolabs, Ipswich, MA, USA), and 8.75 μL of ddH$_2$O. The PCR amplification program was as follows: 98 °C for 2 min; followed by 30 cycles at 98 °C for 15 s, 55 °C for 30 s, and 72 °C for 30 s; and a final extension at 72 °C for 5 min. The specificity of the amplified final product was detected by 1.2% agarose gel electrophoresis.

### 2.4. Amplification Sequencing and Phylogenetic Tree Construction

The PCR-amplified product was detected by gel electrophoresis. The target length fragment (about 436 bp) was extracted by gels, and the concentration was detected using a Qubit 4.0 fluorometer (Thermo Fisher Scientific, Waltham, MA, USA). The library was constructed by equimixing different samples, and high-throughput sequencing was performed on an Illumina MiSeq system (Shanghai Personal Biotechnology Co., Ltd, Shanghai, China). The raw data were processed using Vsearch (v2.13.4_linux_x86_64) (Shanghai Personal Biotechnology Co., Ltd, Shanghai, China), and the specific process was as follows: Firstly, cutadapt (v2.3) (Shanghai Personal Biotechnology Co., Ltd, Shanghai, China) was used to cut primer fragments and discard sequences that did not match the primers. Secondly, after the removal of chimeras, sequences were grouped by operational taxonomic unit (OTU) at 95% similarity [19,28]. Lastly, the BLASTn tool (http://www.ncbi.nlm.nih.gov/BLAST, accessed on 10 August 2021) was used to analyze the representative sequences of each OTU [29]. The reference sequences with the highest similarity to the representative sequences of the main OTUs were retrieved from GenBank (National Center for Biotechnology Information, Bethesda, MD, USA). A phylogenetic tree was then constructed by the neighbor-joining method using MEGA 5 (Mega Limited, Auckland, New Zealand) with 1000 bootstraps. The reliability of the phylogenetic tree topology was evaluated [30,31].

The nucleotide sequences of comammox *Nitrospira amoA* obtained in this study were submitted to the GenBank database with the accession numbers ON130361-ON130389.

### 2.5. Real-Time Quantitative PCR Performed Using QuantStudioTM 6 Flex

Real-time quantitative PCR performed using QuantStudio$^{TM}$ 6 Flex (Thermo Fisher Scientific, Waltham, MA, USA) was used to determine the bacterial copy number of comammox clade A, comammox clade B, ammonia-oxidizing bacteria (AOB), and ammonia-oxidizing archaea (AOA). Four primer pairs, namely CA3771f/C576r [32], CB377f/C576r [32], *amoA*-1F9/*amoA*-2R5 [33], and Arch-*amoA* F/Arch-*amoA* R [34], were used for the quantification of the four aforementioned *amoA* genes, respectively. The primers used for the amplification of the *amoA* genes of comammox clade A, comammox clade B, AOB, and AOA are presented in Table S1.

The PCR mixture contained 0.2 μL forward primers (10 μM), 0.2 μL reverse primers (10 μM), 0.4 μL of ROX Reference Dye II (50×), 5 μL of T5 Fast qPCR Mix (2×), and 1 μL of 10-fold serially diluted DNA template, and the amplification system volume was supplemented to 10 μL with 3.2 μL ddH$_2$O.

The AOA and AOB quantitative PCR amplification procedures were as follows: initial denaturation at 95 °C for 30 s; 35 cycles of denaturation at 95 °C for 10 s, annealing at 53 °C (AOA) or 55 °C (AOB) for 30 s, and extension at 72 °C for 35 s; and final extension at 72 °C for 1 min [4,5]. The comammox clade A quantitative PCR amplification program was as

follows: initial denaturation at 95 °C for 5 min, 35 cycles of annealing at 94 °C for 45 s and extension at 53 °C for 60 s, and final extension at 72 °C for 56 s. The comammox clade B quantitative PCR amplification program was as follows: initial denaturation at 95 °C for 5 min, followed by 45 cycles of annealing at 94 °C for 10 s and extension at 58 °C for 10 s, and ending with a final extension at 72 °C for 13 s [32]. The sediment DNA was extracted and purified, and the purified DNA was ligated with PMD18-T (Special vector for efficient cloning of PCR products (TA Cloning)) to construct a recombinant plasmid. The plasmid containing the target gene was introduced into *Escherichia coli* and cultivated at a constant temperature (37 °C); then, positive clones were picked and plasmids DNA was extracted. The concentration of extracted DNA was tested with a NanoDrop ND-200 system (IMPLEN, Munich, Germany), and the number of gene copies of the plasmid was calculated according to the concentration of plasmid and gene. PCR testing was used to screen for positive clones. The total PCR reaction system volume was 30 μL of 20.8 μL dd $H_2O$, 3 μL Buffer, 2 μL d NTP, 1 μL forward primer (10 μM), 1 μL reverse primer (10 μM), 2 μL DNA plate, 0.2 μL Tap enzyme. The copy number (the number of plasmids contained in each cell) of plasmids was counted after 10-fold serial dilution, and 8 standard samples (concentration range: 10 ng/μL~$10^{-6}$ ng/μL) and DNA samples were amplified. The plasmid criterion for qPCR was that the correlation reaches 0.96–1.10 and $r^2 \geq 0.99$. The experiment was conducted nine times for each sample. The amplification rate of the standard samples was between 90% and 110%, and the correlation values of the standard curves were greater than 0.99. At the same time, a negative control without template DNA was added to detect and eliminate any potential contamination.

### 2.6. Statistical Analysis

R (version 4.02) (The University of Auckland, Auckland, New Zealand) was used to process the data, and Duncan's and ANOVA tests ($p < 0.05$) were used to evaluate the significance of differences in environmental physicochemical indicators and microbial abundance. Microbial abundance was calculated using QuantStudio$^{TM}$ Real-Time PCR software (version 1.2) (Perkin Elmer, Waltham, MA, USA). Diversity indices were calculated using QIIME2 (Gregory Caporaso, Flagstaff, AZ, USA) [35]. RDA (Redundancy Analysis) of environmental parameters and abundance data was performed using Canoco 5 (version 5) (Microcomputer Power, Ithaca, NY, USA). The "Hmisc" and "corrplot" packages in R (version 4.02) were used to perform a Spearman correlation analysis of abundance, diversity. and environmental parameters and to plot the results ($p < 0.05$). The "circlize" package was employed to visualize the dominant OTUs by plotting a chordal graph. The Mantel test was employed to reveal the correlation between environment parameters and the comammox clade ($p < 0.05$), and the results were analyzed using the "ggcor" and "vegan" packages in R. The "SpiecEasi" package was used to plot the contribution rate of dominant OTUs in the comammox clade and the significant correlation between various dominant OTUs ($p < 0.05$).

## 3. Results

### 3.1. Geochemical Characteristics of Overlying Water and Sediments

The average pH of the overlying water in the main stream of the Three Gorges Reservoir area was 8.33 (Supplementary Table S2), the bottom water temperature was 15.76 °C, the dissolved oxygen in the water was 7.57 mg/L, and the water depth of the sampling points ranged from 22.78 m to 75.05 m.

The sediment physicochemical parameters are shown in Table 1. The sediment pH was 7.41–7.8. The N-$NO_2^-$ content ranged from 0.001 to 0.014 mg/kg, with the highest N-$NO_2^-$ content at sampling site M8 and the lowest N-$NO_2^-$ content at M1 (Table 1). The $NO_3^-$ and $NH_4^+$ concentrations at different points along the river showed minimal change. The highest N-$NO_3^-$ content was observed at M3 (8.14 mg/kg), which was 1.64 times that at M4 with the lowest N-$NO_3^-$ content (4.97 mg/kg). The lowest content of N-$NH_4^+$ at M4 (0.46 mg/kg) was one-third of the highest content at M1 (1.38 mg/kg).

Total carbon (TC) varied from 1.08 g/kg at M2 to 1.54 g/kg at M9. The difference in TC content was small (20.32 g/kg~25.79 g/kg). Meanwhile, the difference in sediment moisture content was significant, and the minimum (33.77%) at M2 was about half of the maximum (58.27%) at M9.

**Table 1.** The physicochemical properties of sediments.

| Sampling Sites | Regions | Moisture Content (%) | pH | $NO_2^-$ -N (mg/kg) | $NO_3^-$ -N (mg/kg) | $NH_4^+$ -N (mg/kg) | TN (g/kg) | TC (g/kg) | C:N |
|---|---|---|---|---|---|---|---|---|---|
| M1 | Chong Qing | 48.75 b | 7.6 abc | 0.001 b | 7.14 bc | 1.38 a | 1.46 ab | 25.76 a | 17.66 c |
| M2 | Feng Du | 33.77 d | 7.80 a | 0.002 ± b | 5.62 bc | 0.49 ab | 1.08 d | 21.36 c | 19.86 a |
| M3 | Zhong Xian | 39.76 c | 7.48 bc | 0.007 ± ab | 8.14 a | 0.66 b | 1.27 c | 21.08 c | 16.56 bc |
| M4 | Zhong Xian | 48.29 b | 7.62 abc | 0.004 b | 4.97 c | 0.46 b | 1.42 b | 20.32 c | 14.28 d |
| M5 | Wang Zhou | 50.38 b | 7.52 bc | 0.009 ab | 6.13 bc | 0.51 b | 1.51 ab | 20.92 c | 13.87 d |
| M6 | Feng Jie | 50.27 b | 7.65 ab | 0.006 ab | 6.23 bc | 0.41 b | 1.53 a | 21.64 c | 14.11 d |
| M7 | Wu Shan | 48.77 b | 7.56 bc | 0.018 b | 5.49 b | 0.42 b | 1.47 ab | 20.49 c | 13.99 d |
| M8 | Ba Dong | 52.07 b | 7.41 c | 0.014 a | 5.65 c | 0.57 b | 1.46 ab | 23.39 b | 16.02 c |
| M9 | Zi Gui | 58.27 a | 7.45 bc | 0.007 b | 5.93 bc | 0.94 b | 1.54 a | 21.22 c | 13.82 d |

Notes: One-way ANOVA was applied ($p < 0.05$, Duncan's test). Different lowercase letters in the same column indicate significant differences. MC represents moisture content. TN represents total nitrogen. TC represents total carbon.

### 3.2. Abundance of Comammox Bacteria and Traditional Ammonia-Oxidizing Bacteria

The abundances of comammox *Nitrospira* clade A and clade B *amoA* genes in the main stream of the Three Gorges Reservoir area showed an upward trend along the river, reaching the maximum in the middle part of the reservoir area and decreasing in the tail part of the reservoir area. The highest abundance of clade A and clade B *amoA* genes was $4.50 \times 10^4 \pm 9734.79$ copies/g (M7) and $4.25 \times 10^3 \pm 1050.50$ copies/g (M4), respectively. The lowest abundance of clade A and clade B *amoA* genes was $1.86 \times 10^4 \pm 4582.10$ copies/g (M5) and $4.96 \times 10^2 \pm 350.39$ copies/g (M1), respectively. The abundance of *amoA* genes in AOA was higher in the upper section of the reservoir area (Figure 2) and exhibited a gradual downward trend along the river stream. No significant spatial difference in the abundance of AOB was observed.

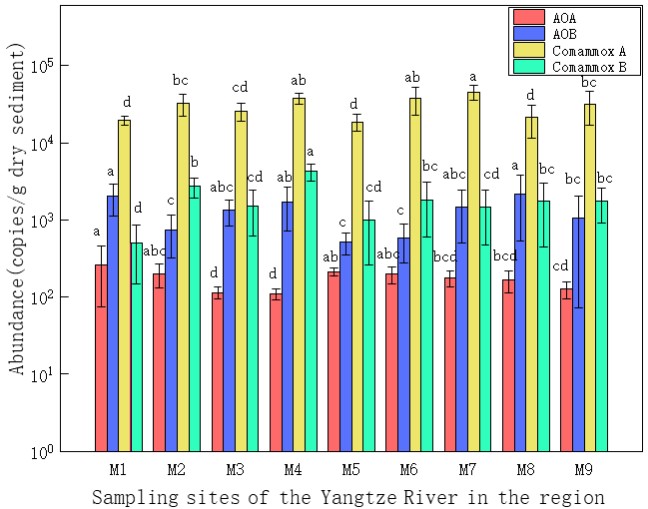

**Figure 2.** Abundance of AOA, AOB, comammox clade A, and comammox clade B *amoA* at the TGR area. Different lowercase letters (a–d) indicate a significant difference among different *amoA* genes at the level of $p < 0.05$ by Duncan's test.

The abundance of comammox *Nitrospira* clade A *amoA* genes was $3.00 \times 10^4 \pm 8782.37$ copies/g, which was significantly higher than that of the other three functional genes. The *amoA* gene abundance of AOA was $1.73 \times 10^2 \pm 48.05$ copies/g, and the *amoA* gene abundance of AOB was $1.28 \times 10^3 \pm 574.69$ copies/g. The *amoA* gene abundance of comammox *Nitrospira*

clade B was $1.83 \times 10^3 \pm 1019.82$ copies/g (Supplementary Figure S1), and that of AOA from each sampling point was the lowest. There was no significant difference in abundance between AOB and comammox *Nitrospira* clade B *amoA* genes (Supplementary Figure S1).

The species richness and Shannon indices exhibited similar spatial variation trends as both indices increased rapidly from M1 and then showed a gradual downward trend (Supplementary Table S3). The Simpson index was the highest at M1 and then showed a downward trend, but it exhibited a high value at M7.

### 3.3. Correlation between Comammox Abundance and Soil Physicochemical Properties

The abundance of comammox clade A genes was positively correlated with pH ($r = 0.47$, $n = 9$), which was highly significantly correlated with $NH_4^+$ ($r = 0.75$, $p < 0.05$, $n = 9$) (Figure 3). The abundance of comammox clade B genes was also positively correlated with pH ($r = 0.4$, $n = 9$). AOA *amoA* gene abundance was negatively correlated with $NO_2^-$ ($r = -0.24$, $n = 9$). AOB *amoA* gene abundance was also negatively correlated with pH ($r = -0.32$, $n = 9$) and TN ($r = -0.33$, $n = 9$). Both AOA and AOB *amoA* gene abundances were positively correlated with TC ($r = 0.22$, $n = 9$) ($p < 0.05$).

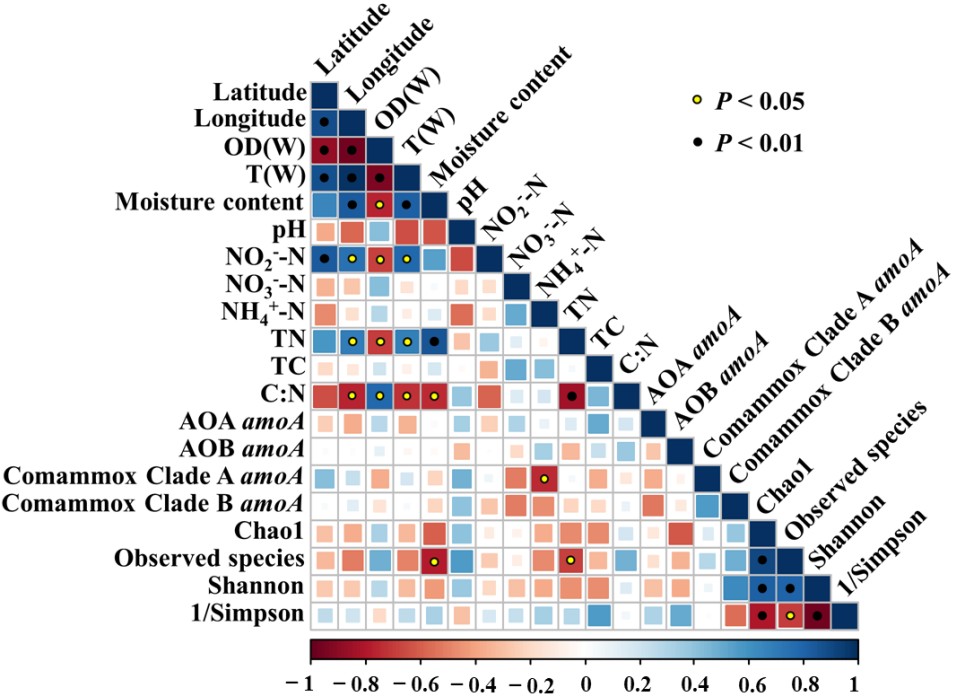

**Figure 3.** Heat map of correlation between AOA, AOB, comammox clade A, comammox clade B *amoA*, and environmental physicochemical indexes. Blue represents a positive correlation; red represents a negative correlation. Only the significant correlations are labeled in circles at the level of $p < 0.05$ by Duncan's test. Scale bars indicate correlation coefficients. This figure sampled R language for analysis.

### 3.4. Phylogenetic Tree of Comammox Bacteria

A total of 727,323 comammox gene sequences were obtained by high-throughput sequencing, with sequences per sample ranging from 56,789 to 101,426 in the community analysis. In total, 29 OTUs with a relatively high abundance were screened (accounting for 65.34% of the comammox *Nitrospira* sequence (the proportion of the reads for 29 selected OTUs out of the total reads)) for phylogenetic analysis. These 29 OTUs fell into three clades: comammox clade A.1 (11 OTUs), comammox clade A.2 (4 OTUs), and comammox clade B (14 OTUs). The dominant OTUs of comammox clade A.1 accounted for 18.69%, followed by comammox clade A.2 (18.58%) and comammox clade B (14.30%) (Figure 4). The dominant

OTU of comammox clade A.1 was OTU24 (3.57%), that of comammox clade A.2 was OTU1 (11.18%), and that of comammox clade B was OTU7 (8.38%) (Figure 4).

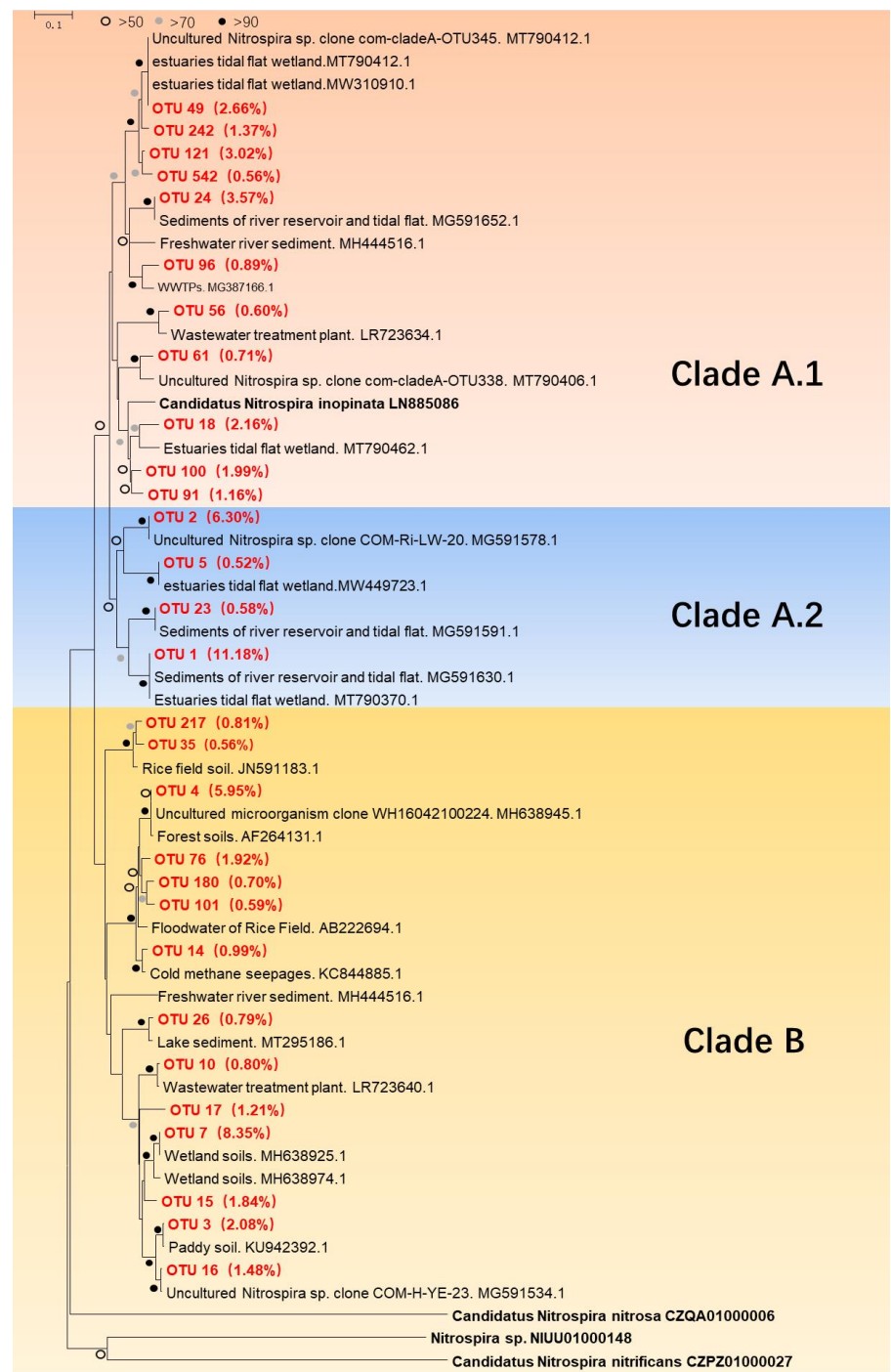

**Figure 4.** Phylogenetic analysis of comammox *amoA* genes. The percentage in parentheses (behind OTU) indicates the proportion of each OTU to the total number of comammox *amoA* genes. Only the obtained OTUs with more than 0.52% comammox *amoA* gene sequence are displayed in the phylogenetic tree. The reference sequence comes from GenBank. The letters and numbers behind the sequence are the submission serial number. The circle, gray dot, and black dot at the clade nodes represent 50~70%, 70~90%, and 90~100% sequence similarity, respectively. The percentages in parentheses next to the OTUs indicate the proportion of each OTU in the total sequence of comammox *amoA* genes. The calculation method was the maximum composite likelihood method. The scale bar represents 20% of the sequence difference.

At sampling point M8, the number of gene sequences was the largest (70,762 sequences). At sampling point M3, the number of gene sequences was the smallest (36,179 sequences). Clades A.1, A.2, and B were most abundant at sampling points M2, M9, and M2, respectively (Supplementary Figure S2).

### 3.5. Interaction between Environment and Comammox Bacteria

Figure 5a mainly expresses the degree of influence of different physical and chemical properties on OTUs of the comammox clades. DO had a significant positive effect on the relative OTU abundance of comammox clade A.2 and comammox clade B communities (Figure 5a) but had no effect on comammox clade A.1. The community of comammox clade A.2 was affected by the environmental $NO_3^-$, which showed a correlation, but the concentration of $NO_3^-$ had little effect on comammox clade B and comammox clade A.1.

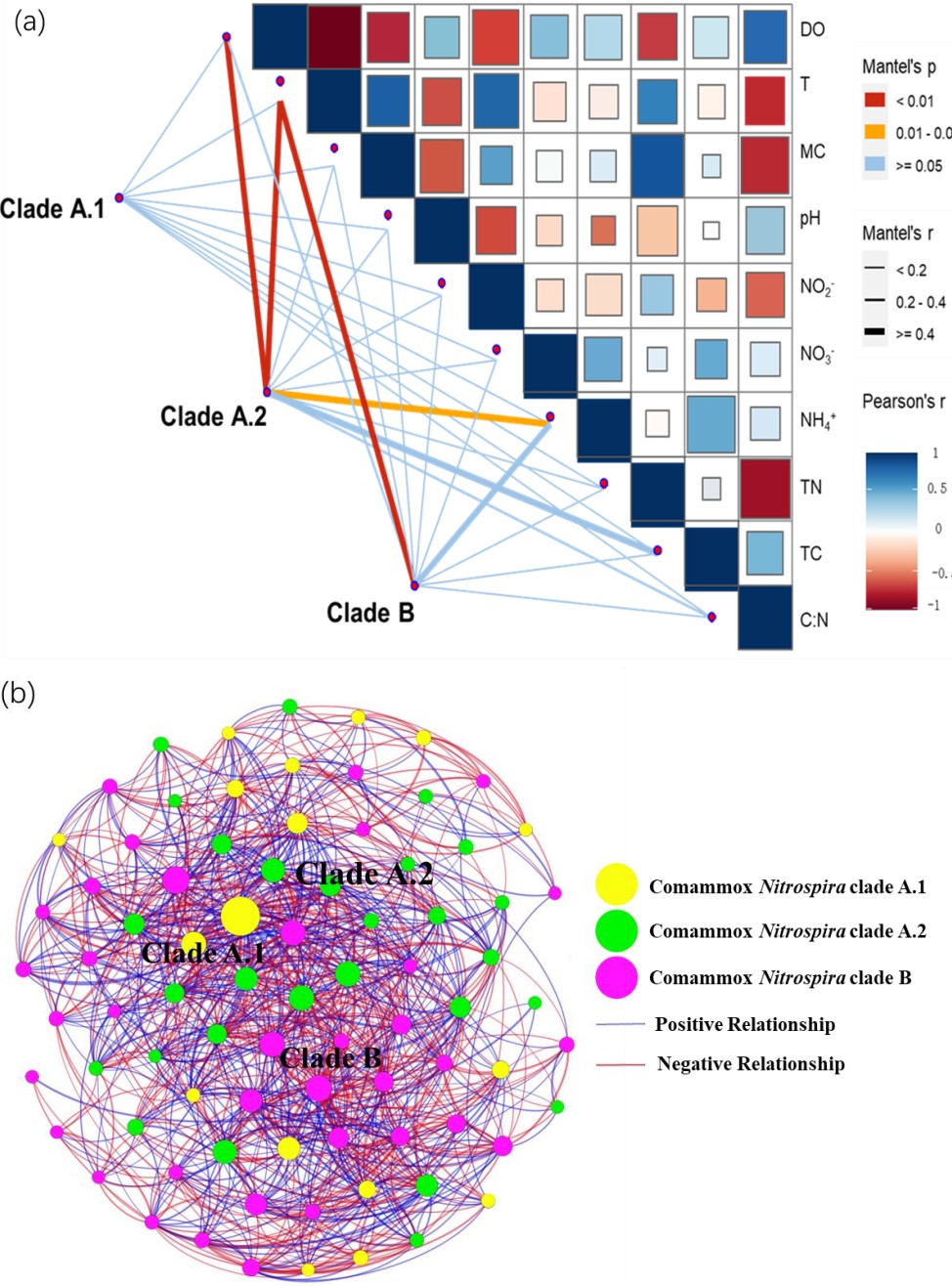

**Figure 5.** (**a**) Correlations between comammox clade A.1, comammox clade A.2, and comammox clade B abundances and environmental physicochemical indicators. (**b**) Contribution analysis of each

clade network of comammox clades A.1, A.2, and B. Each node in the network represents a microbial population. The size of the nodes indicates the degree of connection, and the color of the nodes indicates the different main species. Blue lines represent a positive correlation, whereas red lines represent a negative correlation ($p < 0.05$). The size of the circle represents the specific gravity of the OTU. The numbers below the OTU color bar represent the number of OTUs.

The two axes of the RDA jointly explain 99.60% of the species information change and the species–environment relationship (Supplementary Figure S3). The abundances of *amoA* genes in the two clades of comammox *Nitrospira* showed a significant positive correlation (r = 0.26, $p < 0.05$, $n = 9$). The abundance of comammox *Nitrospira* clade A *amoA* genes was positively correlated with $NO_3^-$ (r = 0.23, $p < 0.05$, $n = 9$) and DO (r = 0.17, $p < 0.05$, $n = 9$). The abundances of comammox *Nitrospira* clade B *amoA* and AOB *amoA* genes were significantly positively correlated with TN (r = 0.01, $p < 0.05$, $n = 9$), TC (r = 0.15, $p < 0.05$, $n = 9$), and pH (r = 0.15, $p < 0.05$, $n = 9$). AOA *amoA* was positively correlated with $NH_4^+$ (r = 0.14, $p < 0.05$, $n = 9$).

As shown in the network plot, there are more connecting lines of positive correlations than negative correlations in the network (Figure 5b) between the OTUs, which indicates that these OTUs may have stronger cooperative mechanisms and weaker competitive mechanisms. The number of OTUs in comammox clades A.1, A.2, and B accounted for 21.78%, 19.80%, and 58.42% of the total OTUs, respectively. In the environment, comammox clade B displayed the highest abundance, whereas comammox clade A.1 exhibited the lowest abundance.

## 4. Discussion

### 4.1. Comammox Nitrospira Clade Differentiation

In different environments, the proportion of each branch of comammox *Nitrospira* differs [7]. In this study, our phylogenetic tree showed that the comammox bacteria in the sample fell into three branches, namely, comammox clade A.1, comammox clade A.2, and comammox clade B. Comammox clade A was more widely distributed than comammox clade B, but the distribution difference was not significant, which might be due to the scouring of the middle and lower reaches of the riverbed and the coarsening of particles, which affects the microbial niche and diversity [19,36]. Consistently, many studies have also found that comammox clade A is more widely distributed than comammox clade B in both artificial and natural ecosystems [14,37,38], and that comammox clade A.1 is more widespread than comammox clade A.2 in both ecosystems [39,40].

There were a lot of OTUs of comammox clade A.1 at sampling site M2, mainly from estuarine wetlands, the river reservoir, tidal flat sediments, freshwater river sediments, and sewage treatment plants [7,41]. Comammox clade A.2 was most abundant at sampling site M9, and it was mainly distributed in estuarine tidal flat wetlands. Comammox clade B accounted for the highest proportion (28.07%) at M2, mainly from paddy soil, forest soil, wetland soil, lake bottom sediments, and wastewater plants [7,24,42,43].

This study found that the abundance of both comammox clade A and comammox clade B ($3.31 \times 10^4$ copies $g^{-1}$ and $2.01 \times 10^3$ copies $g^{-1}$, respectively) in the midstream of the reservoir (M3–M7) was slightly higher than that in the upstream ($2.59 \times 10^4$ copies $g^{-1}$ and $1.58 \times 10^3$ copies $g^{-1}$, respectively) and downstream parts ($3.26 \times 10^4$ copies $g^{-1}$ and $1.75 \times 10^3$ copies $g^{-1}$, respectively), indicating that comammox had better adaptability to the water environment in this section. Some studies have found that after damming, physical changes in rivers will cause sediment deposition [44]. Along the direction of water flow, there is a high deposition rate at the tail of the reservoir, and deposition decreases rapidly with decreasing distance from the dam. Large particles settle down first, and small particles settle down later or have difficulty in settling [21,22]. Dam building also leads to changes in the chemical and biological composition of the reservoir [19,36]. Previous studies have found that in the sediments of the main stream of the Yangtze River, comammox *Nitrospira* have large-scale niche differentiation depending on the altitude, and climate,

topography, and landform are the main influencing factors [38]. Our results indicated that the water environment changes after the construction of the dam may be responsible for the differentiation of comammox bacteria in the entire reservoir area. This study showed there were certain spatial changes in the abundance of comammox along the water flow. However, the DNA extracts were not tested for inhibiting substances in the experiment, which could decrease the detected number of *amoA* copies. Thus, the intensity of the spatial changes may not be fully reflected to some extent.

### 4.2. Coexistence and Differentiation of Comammox and Traditional Ammonia-Oxidizing Bacteria

Comammox, AOA, and AOB have been reported to exist in various environments such as soil and tidal flats [7,45]. Consistently, comammox, AOA, and AOB were detected in this study, indicating that comammox bacteria were widely present in the main stream of the Three Gorges Reservoir area. The abundance of comammox clade A genes was higher than that of AOA and AOB genes in this study area. Our result is similar to a previous study reporting that in coastal waters and river sediments, the abundance of comammox *amoA* genes was higher than that of AOA *amoA* and AOB *amoA* genes [7]. The existing studies have shown that comammox bacteria have a lower nitrous oxide emission rate than AOB, and their emission rate is relatively close to that of AOA [46–49]. Our data indicated that the abundance of clade A genes in the main stream of the Three Gorges Reservoir area was much higher than that of AOA and AOB genes, which might be beneficial to the reduction in nitrous oxide production in the reservoir area.

Most of the existing studies have shown that comammox have a higher transcription rate than AOA and AOB [15] and a higher affinity to ammonia, and thus, they have a competitive advantage in low-ammonia environments. Several studies have found that comammox clade A can adapt to eutrophic conditions [18,21]. Our data showed no significant positive correlation between the abundance of comammox clade A or clade B genes and $NH_4^+$, indicating that the ammonia concentration in the sediment in this study area may be relatively high and has not yet become a limiting factor for the growth of comammox bacteria.

Our results showed that the species richness and Shannon indices were similar in their space variation, and the Simpson index representing species diversity showed a change trend opposite to that of the species richness and Shannon indices. The highest species richness and diversity of comammox were observed in the middle part of the reservoir area, which might be because the relatively stable water environment in the middle of the reservoir area was more conducive to the species reproduction of comammox bacteria. In the upper part of the reservoir area, large particles of sediment in the river continued to settle and cover the original sediment, which may have interfered with the growth of comammox bacteria. However, the continuous discharge of water from the Three Gorges Dam resulted in the rapid flow of the river water in the lower part of the reservoir area. In this way, the sediments near the dam were more susceptible to the rapid flow of water.

### 4.3. Influence of Geographical Environment on Comammox and Traditional Ammonia-Oxidizing Microorganisms

Ammonia is the common substrate of AOA/AOB/comammox bacteria, and these three ammonia-oxidizing bacteria generally compete for ammonia. This study found that $NH_4^+$ concentration negatively correlated with comammox clade A *amoA* gene abundance but positively correlated with AOA *amoA* gene abundance. There was also a negative correlation between $NH_4^+$ concentration and comammox clade A *amoA* gene abundance in tidal flat sediments of the Yangtze River [32]. This study also found no significant correlation between $NH_4^+$ concentration and AOB *amoA* gene abundance. Xu et al. (2020) found that high ammonia concentrations might provide sufficient reaction substrates for aerobic AOB [8]. These findings suggest that the effects of ammonia on these ammonia-oxidizing microorganisms are complex, and that there is not necessarily competition for ammonia among these microorganisms.

pH has long been recognized as a major factor affecting the activity and distribution of ammonia-oxidizing microorganisms in ecosystems [50]. Some sediment and soil studies have found that comammox clade A prefers high-pH environments [8,12]. In soil studies, pH has been reported to be an important factor affecting the niche differentiation of AOA and AOB [51]. In this study, pH was positively correlated with comammox clade B *amoA* and AOB but exhibited no significant correlation with the abundance of AOA *amoA* genes and comammox clade A *amoA* genes. The pH, T, and $NO_2^-$ had more significant effects on the niche differentiation of comammox clade A and comammox clade B than on that of AOA and AOB.

## 5. Conclusions

In this study, we studied comammox in river sediments in the TGR area, and comammox clade A, comammox clade B, AOA, and AOB were detected in all the samples. The abundance of comammox clade A *amoA* genes was higher than that of AOA *amoA* genes and AOB *amoA* genes, and comammox *Nitrospira* was the dominant species among the ammonia-oxidizing microorganisms. The abundance of comammox *Nitrospira* clade A and clade B *amoA* genes in the main stream of the Three Gorges Reservoir showed an upward trend along the river stream and reached the maximum in the middle part of the reservoir area, whereas the highest AOA *amoA* gene abundance value appeared in the upper part of the reservoir area. The abundance of comammox *Nitrospira* clade A was the highest ($3.00 \times 10^4 \pm 8782.37$ copies/g), followed by comammox *Nitrospira* clade B, AOB, and AOA. Comammox clade B had the highest percentage, followed by comammox clade A.1 and comammox clade A.2. Changes in environmental parameters will cause the niche differentiation of comammox bacteria, which may change their ammonia oxidation process in the sediment of the TGR area. DO in overlying water significantly correlated with the relative abundance of OTUs for comammox clade A.2 and clade B. Furthermore, DO, $NO_3^-$, and TC had significant effects on comammox clade A, while pH had significant effects on comammox clade B. This study confirms that the construction of the TGR had an impact on the niche differentiation of comammox bacteria in the main stream of the reservoir area.

**Supplementary Materials:** The following supporting information can be downloaded at: https://www.mdpi.com/article/10.3390/w14244014/s1, Table S1: Primer information table, Table S2: Physiochemical properties of Overlying water, Table S3: Alpha diversity index table, Figure S1: Total abundance of AOA, AOB, comammox clade A and comammox clade B amoA in samples, Figure S2: The distribution of 29 dominant OTUs at different sampling sites. The 9 sampling points and 29 OTUs are represented by different colors, Figure S3: Redundancy analysis of comammox cladeA, comammox clade B, AOA and AOB amoA abundances at sampling sites.

**Author Contributions:** Conceptualization, M.H. and J.Z. (Jianwei Zhao); data curation, S.L. (Shuang Liu); formal analysis, S.L. (Shuang Liu); funding acquisition, Y.W. and J.Z. (Jianwei Zhao); investigation, J.Z. (Jiahui Zhang), M.H. and J.W.; methodology, J.Z. (Jiahui Zhang) and J.W.; project administration, M.H., Y.W., Y.B. and J.Z. (Jianwei Zhao); supervision, Y.W.; visualization, Y.B. and J.Z. (Jianwei Zhao); writing—original draft, S.L. (Shuang Liu); writing—review and editing, S.L. (Shanze Li) and J.Z. (Jianwei Zhao). All authors have read and agreed to the published version of the manuscript.

**Funding:** This study was supported by the National Natural Science Foundation of China (No. U2040211, No. U1802241, No. 92047204, and No. 92047203), the National Key Research and Development Project (2021YFC3201002), the project of the China Three Gorges Corporation (No. 201903144), the China Institute of Water Resources and Hydropower Research (SKL2020TS07), and Follow-up Work of the Three Gorges Project (2136902).

**Institutional Review Board Statement:** Not applicable.

**Informed Consent Statement:** Not applicable.

**Data Availability Statement:** The nucleotide sequences of comammox *Nitrospira amoA* gene obtained in this study are available in GenBank under accession number(s) ON130361-ON130389.

**Acknowledgments:** The authors express gratitude to the linguistics Ping Liu from Huazhong Agriculture University, Wuhan, China, for her work at English editing and language polishing.

**Conflicts of Interest:** The authors declare no conflict of interest.

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
