# Peer review of "Effects of Dam Building on Niche Differentiation of Comammox Nitrospira in the Main Stream of the Three Gorges Reservoir Area"

_water, doi:10.3390/w14244014_

Round 1
Reviewer 1 Report
The authors attempt to describe the impact of a dam on niche differentiation of comammox bacteria in the Three Gorges Area. They present evidence of variation at different sampling points along the river/reservoir but more detailed site information is needed to specifically evaluate the effects of dam building. There is a lot of good microbial ecology information but additional work is needed on the presentation and interpretation.
1) Lines 21-23: The discussion of OTU results is very confusing and should be re-written.
2) Line 31: Even though DO may have correlated, it doesn’t necessarily indicate causation.
3) Lines 33-34: Seems like an overly strong conclusion. How do you eliminate the possibility of natural variation along the river or other causes besides the dam itself, such as changes in land use?
4) Line 100: Should add detailed information on each sampling site. The authors discuss points above and in the reservoir throughout the manuscript, but it was hard for me to know where the distinctions between these sites were.
5) Figure 1: It is very difficult to see the sampling points and dam on the map.
6) Lines 114: What does “remove to constant temperature” mean?
7) Lines 114-120: Should add more details on how these analyses were done or add references. For example, was the pH measured on a 1:1 mixture with water or some other way?
8) Lines 136-150: This section needs to be re-written. The sequencing details and initial bioinformatics quality control methods are missing.
9) Lines 137: The authors used 95% for the OTU cutoff. This seems very high for a functional gene like amoA. Is there a reference to support using this OTU value?
10) Line 174: It seems that plasmids were used for the standard curves. Add details on where these came from and how they were used.
11) Lines 179: Did the authors test the samples for PCR inhibition?
12) Lines 181-193: Need to add Mantel test to methods.
13) Line 200: Table 1 instead of Table 2?
14) Table 1: Does MC equal % moisture content?
15) Table 1: Table is very difficult to read with all of the numbers. Suggest removing +/- values since statistics letters are provided.
16) Figure 2: Move Figure 2b to supplemental. It repeats info in Figure 2a. Increase size of Figure 2a.
17) Line 260: Seems more informative to show this as % of the total reads rather than % of total OTUs.
18) Lines 262-269: This is too little description for figures that cover 2 pages. Need to move Figure 4a or 4b to supplemental and/or spend more text walking the reader through the important points of these figures.
19) Lines 297-302: I don’t follow this material. Can you clarify what is meant by “…clade A.1 contributed 24.88% of all samples…”?
20) Figure 5: Can move Fig 5a or 5b to supplemental since they are very similar analyses.
21) Line 330: Need more info to know where the middle of the reservoir is.
22) Lines 340-342: Not sure how the presented results directly say anything about sediment subsidence.
23) Line 409: Needs to be re-phrased since the authors did not test whether comammox bacteria directly oxidized ammonia to nitrate.
24) Lines 416-419: Too strong of conclusion based on correlation
Author Response
The authors attempt to describe the impact of a dam on niche differentiation of comammox bacteria in the Three Gorges Area. They present evidence of variation at different sampling points along the river/reservoir but more detailed site information is needed to specifically evaluate the effects of dam building. There is a lot of good microbial ecology information but additional work is needed on the presentation and interpretation.
1) Lines 21-23: The discussion of OTU results is very confusing and should be re-written.
Answer: Changed “and the OTUs detected from comammox clade B (14 OTUs) were more than those detected from comammox clade A. 1 (11 OTUs) and comammox clade A.2 (4 OTUs)” to “while comammox clade B had more OTUs (14 OTUs) than comammox clade A.1 (11 OTUs) and comammox clade A.2 (4 OTUs)”
2) Line 31: Even though DO may have correlated, it doesn’t necessarily indicate causation.
Answer: Changed “The results indicate that overlying water dissolved oxygen (DO) in the reservoir area had a significant effect on the differentiation of comammox clade A into comammox clade A.1 and comammox clade A.2.” to “Additionally, the relative abundances of both comammox clade A.2 and clade B were significantly correlated with overlying water dissolved oxygen (DO) in the reservoir area”.
3) Lines 33-34: Seems like an overly strong conclusion. How do you eliminate the possibility of natural variation along the river or other causes besides the dam itself, such as changes in land use?
Answer: Thanks for reviewers for this comment. We checked this conclusion. As the reviewer said, it maybe seems like an overly strong conclusion. We improved this conclusion, changed “This study confirms that the construction of the Three Gorges Dam has an impact on the niche differentiation of comammox Nitrospira in the main stream of the Three Gorges Reservoir area.” to “This study thus indicated that there exists a niche differentiation of comammox Nitrospira in the main stream of the Three Gorges Reservoir area.”
4) Line 100: Should add detailed information on each sampling site. The authors discuss points above and in the reservoir throughout the manuscript, but it was hard for me to know where the distinctions between these sites were.
Answer: Thanks for the reviewer's suggestions. We arranged one sampling site in each county to make the sampling site layout more scientific and reasonable. At the same time, we also hope to find some differences between counties. But the results do not prove that there is a significant difference between these counties. We added the corresponding details to the manuscript: “There are nine administrative regions from the beginning to the end of the reservoir, and one or two sampling points were arranged in each region depending on the area of the region (M1-9).
5) Figure 1: It is very difficult to see the sampling points and dam on the map.
Answer:In order to see the sampling point and the dam clearly, the color of the sampling point has been changed to white and the dam to red.
6) Lines 114: What does “remove to constant temperature” mean?
Answer:We improved this sentence. Changed “The fresh sediment was dried at 105°C remove to a constant temperature to measure the moisture content” to “The fresh sediment was dried at 105 °C to a constant weight to determine the moisture content”
7) Lines 114-120: Should add more details on how these analyses were done or add references. For example, was the pH measured on a 1:1 mixture with water or some other way?
Answer: Thanks for the reviewer's suggestions. We provide more detailed test information. The specific improvements are as follows:
Change “The sediment pH was determined with a pH meter (METTLER TOLEDO, Switzerland)” to: “The sediment pH was determined using a pH meter (Mettler Toledo, Zurich, Switzerland) at a sediment/water suspension ratio of 1:2.5 (w/v)”
Change “The air-dried ground sediment was leached with 2 mol/L KCl, and then ammonia nitrogen (NH4+-N), nitrate nitrogen (NO3--N), and nitrite nitrogen (NO2--N) were determined by a flow analyzer (SEAL Analytical GmbH, Norderstedt, Germany). ” to: “Ten grams of air-dried sediment was leached with 50 mL 2 mol/L KCl, and then, the ammonia nitrogen (NH4+-N), nitrate nitrogen (NO3--N), and nitrite nitrogen (NO2--N) levels of the filtrate were determined using a flow analyzer (Seal Analytical GmbH, Norderstedt, Germany)”
8) Lines 136-150: This section needs to be re-written. The sequencing details and initial bioinformatics quality control methods are missing.
Answer: Thanks for the reviewer's suggestions. We added some measurement details and rewrote this section. The revised content is as follows: “The PCR-amplified product was detected by gel electrophoresis. The target length fragment (about 436 bp) was extracted by gels, and the concentration was detected using a Qubit 4.0 fluorometer (Thermo Fisher Scientific, Massachusetts, USA). The library was constructed by equimixing different samples, and high-throughput sequencing was performed on an Illumina MiSeq system (Shanghai Personal Biotechnology Co., Ltd). The raw data were processed using Vsearch (v2.13.4_linux_x86_64), and the specific process was as follows: Firstly, cutadapt (v2.3) was used to cut primer fragments and discard sequences that did not match the primers. Secondly, after the removal of chimeras, sequences were grouped by operational taxonomic unit (OTU) at 95% similarity [28,29]. Lastly, the BLASTn tool (http://www.ncbi.nlm.nih.gov/BLAST) was used to analyze the representative sequences of each OTU [30]. The reference sequences with the highest similarity to the representative sequences of the main OTUs were retrieved from GenBank. A phylogenetic tree was then constructed by the neighbor-joining method using MEGA 5 with 1000 bootstraps. The reliability of the phylogenetic tree topology was evaluated [31,32].”
9) Lines 137: The authors used 95% for the OTU cutoff. This seems very high for a functional gene like amoA. Is there a reference to support using this OTU value?
Answer: Thank reviewer for this comment. We have repeatedly checked the latest studies of comammox. In recent studies, most of them have taken 95% as the cut-off value of OTU. Referring to these studies, we took 95% as the cut-off value of this study. We added these references to the manuscript. The specific references are:
28 Schloss, P.D. Secondary structure improves OTU assignments of 16S rRNA gene sequences. Isme J 2013, 7, 457-460, doi:10.1038/ismej.2012.102.
- Zhao, Z.; Huang, G.; He, S.; Zhou, N.; Wang, M.; Dang, C.; Wang, J.; Zheng, M. Abundance and community composition of comammox bacteria in different ecosystems by a universal primer set. Sci Total Environ 2019, 691, 146-155, doi:10.1016/j.scitotenv.2019.07.131.
10) Line 174: It seems that plasmids were used for the standard curves. Add details on where these came from and how they were used.
Answer: Thanks for the reviewer's suggestions. We have added some measurement details and improved the measurement process. We added “The plasmid containing the target gene was introduced into Escherichia coli and cultivated at a constant temperature (32 ℃). Then, the plasmid DNA was extracted. The concentration of extracted DNA was tested with a NanoDrop ND-200 system (IMPLEN, Munich, Germany), and the number of gene copies of the plasmid was calculated according to the concentration of plasmid and gene.”
11) Lines 179: Did the authors test the samples for PCR inhibition?
Answer: Yes, PCR inhibition was made. We supplemented relevant information, add: “At the same time, a negative control without template DNA was added to detect and eliminate any potential contamination.”
12) Lines 181-193: Need to add Mantel test to methods.
Answer: We added Mantel test information. Changed “and the "ggcor" package was employed to plot significant correlation between the comammox clade and each physicochemical index (p < 0.05)” to “The Mantel test was employed to reveal the correlation between environment parameters and the comammox clade (p < 0.05), and the results were analyzed using the “ggcor” and “vegan” packages in R.”
13) Line 200: Table 1 instead of Table 2?
Answer: We are sorry that the quotation here is incorrect. Changed “Table 2” to “Table 1”
14) Table 1: Does MC equal % moisture content?
Answer: MC is moisture content. We apologize for not being able to write the full name. We have improved this indicator, changed “MC” to “Moisture content (%)”
15) Table 1: Table is very difficult to read with all of the numbers. Suggest removing +/- values since statistics letters are provided.
Answer: We improved the table according to the comments. +/- values were deleted.
16) Figure 2: Move Figure 2b to supplemental. It repeats info in Figure 2a. Increase size of Figure 2a.
Answer: Figure 2b has been move to supplemental and the size of Figure 2a has been increased.
17) Line 260: Seems more informative to show this as % of the total reads rather than % of total OTUs.
Answer: We apologize for not elaborating accurately. We improved the sentence. Changed “65.34% means that 29 OTUs are selected from all samples and account for 65.34% of the total OTU)” to “the proportion of the reads for 29 selected OTUs out of the total reads ”
18) Lines 262-269: This is too little description for figures that cover 2 pages. Ned to move Figure 4a or 4b to supplemental and/or spend more text walking the reader through the important points of these figures.
Answer: Thanks for the reviewer's suggestions. We feel that Figure 4a is important and needs to be retained in the text. However, Figure 4b is not very important, and the information of this figure is also limited. Therefore, according to the suggestions of the reviewer, we used Figure 4b as supplementary material.
19) Lines 297-302: I don’t follow this material. Can you clarify what is meant by “…clade A.1 contributed 24.88% of all samples…”?
Answer: All samples here refer to quantity of all OTUs. We apologize for the inaccuracy. We improved these sentences. Changed “Comammox clade A.1 contributed 24.88% of all samples, comammox clade A.2 contributed 33.15% of all samples, and comammox clade B contributed the highest percentage of all samples at 45.98%. In the environment, comammox clade B displayed the highest abundance, whereas comammox clade A.1 exhibited the lowest abundance” to “The number of OTUs in comammox clades A.1, A.2, and B accounted for 24.88%, 33.15%, and 45.98% of the total OTUs, respectively.”
20) Figure 5: Can move Fig 5a or 5b to supplemental since they are very similar analyses.
Answer: Done. We moved Fig. 5b to the supplementary material.
21) Line 330: Need more info to know where the middle of the reservoir is.
Answer: We supplemented the abundance information of various parts in the reservoir area. Changed “This study found that the abundance of comammox clade A and comammox clade B was higher in the middle section of the reservoir area” to “This study found that the abundance of comammox clade A and comammox clade B (3.31 × 104 copies g-1 and 2.01 × 103 copies g-1, respectively) in the midstream of the reservoir (M3-M7) was higher than that in the upstream (2.59 × 104 copies g-1 and 1.58 × 103 copies g-1, respectively) and downstream parts (3.26 × 104 copies g-1 and 1.75 × 103 copies g-1, respectively).”
22) Lines 340-342: Not sure how the presented results directly say anything about sediment subsidence.
Answer: Thank the reviewers for their valuable comments. After the dam is built, the sedimentation process usually changes from the head to the end of the reservoir. However, in this paper, we have not measured the detailed changes of these sediments. So, we rewrote this sentence to make it more rigorous. Changed “Our results showed that the sediment subsidence in the Three Gorges Reservoir area caused by the dam construction and the resultant water environment changes may be mainly responsible for the differentiation of comammox bacteria in the entire reservoir area.” to “Our results indicated that the water environment changes after the construction of the dam may be responsible for the differentiation of comammox bacteria in the entire reser-voir area.”
23) Line 409: Needs to be re-phrased since the authors did not test whether comammox bacteria directly oxidized ammonia to nitrate.
Answer: According to literature reports, comammox bacteria can directly oxidize ammonia nitrogen to nitrate. It can also reduce the production of nitrous oxide during ammonia oxidation compared to ammonia oxidizing bacteria. Although these are reported in the literature, they are not the results of this research, and they are not closely related to other conclusions. After much deliberation, we decided to delete these contents to make the conclusion more rigorous. Delete “Comammox bacteria were able to directly oxidize ammonia to nitrate, which completely changed the process of ammonia oxidation. At the same time, comammox bacteria can also reduce the production of nitrous oxide in the process of ammonia oxidation compared with ammonia oxidizing bacteria. These environmental effects profoundly affect the nitrogen cycle process in nature.”
24) Lines 416-419: Too strong of conclusion based on correlation
Answer: We have improved these conclusions to make them more accurate and reasonable. Changed “The change of environmental parameters will cause the niche differentiation of comammox bacteria, which will inevitably change its ammonia oxidation process in the sediment, thereby changing the nitrogen transformation process in the water body of the TGR area. DO in overlying water had a significant effect on the differentiation of comammox clade A into comammox clade A.1 and comammox clade A.2.” to “Changes in environmental parameters will cause the niche differentiation of comammox bacteria, which may change their ammonia oxidation process in the sediment of the TGR area. DO in overlying water significantly correlated with the relative abundance of OTUs for comammox clade A.2 and clade B”
Other improvement: We invited the grammar institutions recommended by the Journal to improve the grammar of our manuscript. The grammar improvement proof is provided.

Reviewer 2 Report
Article water-2037783
Effects of dam building on niche differentiation of comammox Nitrospira in the main stream of the Three Gorges Reservoir area.
The manuscript presents the results of the impact of the construction of the Three Gorges Dam on the differentiation of comammox Nitrospira and other microorganisms. The results are interesting and well-documented.
The obtained results indicate that the abundance of clade A gene in the mainstream of the Three Gorges Reservoir area is much higher than that of AOA and AOB, which might be beneficial to the reduction in the nitrous oxide production in the reservoir area.
In addition, these resultss suggest that the effects of ammonia on the ammonia- oxidizing microorganisms are complex, and that there is not necessarily a competition for ammonia among these microorganisms.
The pH level has been recognized as a major factor affecting the activity and distribution of ammonia-oxidizing microorganisms in studied ecosystems.
However, several points in the manuscript should be corrected and better explained.
The most important:
1) The lack of explanation of Nitrospira sp.- their location in taxonomy, the role of A nad B comammox in the nitrification process. It should be presented after the first section of Introduction.
2) In-depth conclusions are needed -future studies? perspectives? How this knowledge can contribute to functioning of this area?
Other points:
Remember about the sapaces between the text and the citation or other brackets, e.g. L: 39, 45,46, 50, 60 etc in all manuscript.
L103: YSI water quality meter – please, add company, city, state
L115-116: pH meter – the same
L120: elemental analyzer – the same
L128-129: PCR amplification system – the same
L130: DNA Polymerase – the same
L188-193, 212: Statistical analysis – p (small font)
L211: Table abbreviation- The physiochemical…
Refferences: the spelling of journal names should be uniformed
Author Response
Effects of dam building on niche differentiation of comammox Nitrospira in the main stream of the Three Gorges Reservoir area.
The manuscript presents the results of the impact of the construction of the Three Gorges Dam on the differentiation of comammox Nitrospira and other microorganisms. The results are interesting and well-documented.
The obtained results indicate that the abundance of clade A gene in the mainstream of the Three Gorges Reservoir area is much higher than that of AOA and AOB, which might be beneficial to the reduction in the nitrous oxide production in the reservoir area.
In addition, these results suggest that the effects of ammonia on the ammonia- oxidizing microorganisms are complex, and that there is not necessarily a competition for ammonia among these microorganisms.
The pH level has been recognized as a major factor affecting the activity and distribution of ammonia-oxidizing microorganisms in studied ecosystems.
However, several points in the manuscript should be corrected and better explained.
The most important:
- The lack of explanation of Nitrospira- their location in taxonomy, the role of A nad B comammox in the nitrification process. It should be presented after the first section of Introduction.”
Answer: Thanks for the reviewer's suggestion. We added relevant content in the introduction: “All the comammox bacteria found so far affiliated to Nitrospira Lineage II and they can be divided into clade A and clade B. Both clade A and clade B can realize ammonia oxidation. However, comammox clade A is usually dominant in aquatic environments [6], while clade B has advantages in forest soil environments [7].”
- Xia, F.; Wang, J.G.; Zhu, T.; Zou, B.; Rhee, S.K.; Quan, Z.X. Ubiquity and Diversity of Complete Ammonia Oxidizers (Comammox). Applied and Environmental Microbiology 2018, 84, doi:10.1128/aem.01390-18.
7 Pjevac P, Schauberger C, Poghosyan L, Herbold CW, van Kessel MAHJ, Daebeler A, Steinberger M, Jetten MSM, Lucker S, Wagner M, Daims H. 2017. AmoA-targeted polymerase chain reaction primers for the specific detection and quantification of comammox Nitrospira in the environment. Front Microbiol 8: e1508.
In-depth conclusions are needed -future studies? perspectives? How this knowledge can contribute to functioning of this area?
Other comments in the manuscript:
Remember about the sapaces between the text and the citation or other brackets, e.g L: 39, 45,46, 50, 60 etc in all manuscript.
Answer:Done
L103: YSI water quality meter – please, add company, city, state
Answer: Add “EXO2, US YSI, Ohio, USA”
L115-116: pH meter – the same
Answer: Changed “METTLER TOLEDO, Switzerland” to “Mettler Toledo, Zurich, Switzerland”
L120: elemental analyzer – the same
Answer: Changed “Elementar Vario PYRO cube, Germany” to” Elementar Vario PYRO cube,Hanau, Germany”
L128-129: PCR amplification system – the same
Answer: Changed “The 25 μL of the total PCR amplification system contained 2 μL of template DNA” to “The total PCR amplification system (25 μL) (Applied Biosystems, Analytik Jena) contained 2 μL of template DNA”
L130: DNA Polymerase – the same
Answer: Add “New England Biolabs, USA” Changed “5 μL of 5×GC buffer, 0.25 μL of Q5® High-Fidelity DNA Polymerase (New England Biolabs, USA)” to “5 μL of 5×GC buffer, 0.25 μL of Q5® High-Fidelity DNA Polymerase (New England Biolabs, New Jersey, USA)”
L188-193, 212: Statistical analysis – p (small font)
Answer: Done
L211: Table abbreviation- The physiochemical…
Answer: We added the full names of these indicators below the table.
Refferences: the spelling of journal names should be uniformed
Answer: Done
Other improvement: We invited the grammar institutions recommended by the Journal to improve the grammar of our manuscript. The grammar improvement proof is provided.

Round 2
Reviewer 1 Report
The authors have addressed many of the initial comments. Below are some remaining items and additional issues noticed when reading the revised manuscript.
1) Lines 21-22: It seems these numbers represent only selected dominant OTUs. If so, the authors should state this or better yet replace it with total numbers like in lines 318-319.
2) Lines 50-51: The statement about clade B having advantages in forest soil environments is too specific/conclusive. They have been found in numerous sample types.
3) Line 63: Sentence syntax is off.
4) Line 121: Indicate in caption what M1-9 represent.
5) Lines 174-175: List concentration of qPCR primers and reagents used.
6) Lines 185-187: Need to discuss how this standard plasmid was developed. Did the authors clone an environmental PCR amplicon? If so, add the PCR and cloning details.
7) Lines 196-197: It seems that the authors did not test for the presence of any inhibiting substances in the extracts, which could suppress amoA numbers determined by qPCR. This would typically be done by spiking each sample with a known concentration of a target gene and seeing if similar results were obtained between each sample and a spiked clean lab water sample containing the same concentration of the target gene. If this was not done, the authors need to mention this in the discussion and indicate that it could artificially decrease the detected number of amoA copies.
8) Lines 286-287: Seems like an incredibly large number of OTUs for only 56,789 to 101,426 sequences per sample. Basically every 1-2 reads is/are its own OTU???
9) Lines 348-351: The numbers seem very similar based on variability shown in Figure 2 and lack of consistent trend in midstream samples. Did the authors run stats on the numbers grouped by up-, mid-, and down-stream locations? I doubt they would be significantly different.
10) Line 429: Is the +/- value correct? Seems like it is off based on value in Line 245.
Author Response
1) Lines 21-22: It seems these numbers represent only selected dominant OTUs. If so, the authors should state this or better yet replace it with total numbers like in lines 318-319.
Answer:Thanks for the reviewer's comment. Changed “Comammox clade A and comammox clade B were detected in all samples, although comammox clade A was dominant, while comammox clade B had more OTUs (14 OTUs) than comammox clade A.1 (11 OTUs) and comammox clade A.2 (4 OTUs)” to “Comammox clade A and clade B were detected in all samples, and comammox clade A was dominant. The number of dominant OTUs in comammox clade A.1 accounted for 18.69% of the total number of OTUs, followed by comammox clade A.2 (18.58%) and clade B (14.30%).”.
2) Lines 50-51: The statement about clade B having advantages in forest soil environments is too specific/conclusive. They have been found in numerous sample types.
Answer: Thanks for the reviewer's guidance. We improved the sentence. Changed “while clade B has advantages in forest soil environments [7]” to “while clade B has advantages in forest, cropland and other environments [7,8]”.
[8] Wang, Z.; Cao, Y.; Zhu-Barker, X.; Nicol, G.W.; Wright, A.L.; Jia, Z.; Jiang, X. Comammox Nitrospira clade B contributes to nitrification in soil. Soil Biology and Biochemistry. 2019, 135, 392-395, doi: 10.1016/j.soilbio.2019.06.004.
3) Line 63: Sentence syntax is off.
Answer:We apologized for the syntax error. Changed “A recent study has shown that not only could clade A and clade B be differentiated in farmland soil and tidal flats in the Yangtze River estuary, but clade A was also differentiated into three sub-clades: A.1, A.2, and A.3 [7,12-14].”to “A recent study has shown that clade A and clade B could differentiate in farmland soil and tidal flats in the Yangtze River estuary, furthermore, clade A could differentiate into three sub-clades: A.1, A.2, and A.3 [7,13-15].”
4) Line 121: Indicate in caption what M1-9 represent.
Answer: We've added that in the caption. Added “M1-9 represents 9 sampling points in the main stream of the reservoir.”
5) Lines 174-175: List concentration of qPCR primers and reagents used.
Answer: We added concentration of these solutions. Changed “The PCR mixture contained 0.2 μL forward primers, 0.2 μL reverse primers, 0.4 μL ROX, 5 μL of T5 Fast qPCR Mix, and 1 μL of 10-fold serially diluted DNA template, and the amplification system volume was supplemented to 10 μL with 3.2 μL ddH2O.”to “The PCR mixture contained 0.2 μL forward primers (10 μM), 0.2 μL reverse primers (10 μM), 0.4 μL of ROX Reference Dye II (50 ×), 5 μL of T5 Fast qPCR Mix (2 ×), and 1 μL of 10-fold serially diluted DNA template, and the amplification system volume was supplemented to 10 μL with 3.2 μL ddH2O.”
6) Lines 185-187: Need to discuss how this standard plasmid was developed. Did the authors clone an environmental PCR amplicon? If so, add the PCR and cloning details.
Answer:Thanks for the reviewer's opinion. Yes, we cloned an environmental PCR amplicon. We added the PCR and cloning details. Changed “The plasmid containing the target gene was introduced into Escherichia coli and cultivated at a constant temperature (37 ℃). Then, the plasmid DNA was extracted. The concentration of extracted DNA was tested with a NanoDrop ND-200 system (IMPLEN, Munich, Germany), and the number of gene copies of the plasmid was calculated according to the concentration of plasmid and gene” to “The sediment DNA was extracted and purified, and the purified DNA was ligated with PMD18-T to construct a recombinant plasmid. The plasmid containing the target gene was introduced into Escherichia coli and cultivated at a constant temperature (37 ℃), then positive clones were picked and plasmids DNA were extracted. The concentration of extracted DNA was tested with a NanoDrop ND-200 system (IMPLEN, Munich, Germany), and the number of gene copies of the plasmid was calculated according to the concentration of plasmid and gene. PCR testing was used to screen for positive clones. The total PCR reaction system volume was 30 μL of 20.8 μL dd H2O, 3 μL Buffer, 2 μL d NTP, 1 μL forward primer (10 μM), 1 μL reverse primer (10 μM), 2 μL DNA plate, 0.2 μL Tap enzyme.
7) Lines 196-197: It seems that the authors did not test for the presence of any inhibiting substances in the extracts, which could suppress amoA numbers determined by qPCR. This would typically be done by spiking each sample with a known concentration of a target gene and seeing if similar results were obtained between each sample and a spiked clean lab water sample containing the same concentration of the target gene. If this was not done, the authors need to mention this in the discussion and indicate that it could artificially decrease the detected number of amoA copies.
Answer: Thanks for the reviewer's guidance. We are sorry we didn't test inhibiting substances in the extracts. We add a discussion of this deficiency to Part 4.1. At the same time, we made some explanations according to the content of the context: "This study showed there were certain spatial changes in the abundance of comammox along the water flow. However, the DNA extracts were not tested for inhibiting substances in the experiment, which could decrease the detected number of amoA copies. Thus, the intensity of the spatial changes may not be fully reflected to some extent."(End of paragraph 3 in Part 4.1)
8) Lines 286-287: Seems like an incredibly large number of OTUs for only 56,789 to 101,426 sequences per sample. Basically every 1-2 reads is/are its own OTU???
Answer: We apologize for the incorrect value in parentheses. At first, we wrote the number of sequences in parentheses, but later we wanted to try the number of OTUs. In the course of repeated revisions, there was an error in this place. Again, we apologize. Changed “At sampling point M8, the number of OTUs was the largest (70762 OTUs). At sampling point M3, the number of OTUs was the smallest (36179 OTUs)” to” At sampling point M8, the number of gene sequences was the largest (70762 sequences). At sampling point M3, the number of gene sequences was the smallest (36179 sequences)”.
9) Lines 348-351: The numbers seem very similar based on variability shown in Figure 2 and lack of consistent trend in midstream samples. Did the authors run stats on the numbers grouped by up-, mid-, and down-stream locations? I doubt they would be significantly different.
Answer: We have conducted a statistical test on these data of up-, mid-, and down-stream locations, and there is no significant difference between them. We have improved the sentence to make it more accurate.
Changed “This study found that the abundance of comammox clade A and comammox clade B (3.31 × 104 copies g-1 and 2.01 × 103 copies g-1, respectively) in the midstream of the reservoir (M3-M7) was higher than that in the upstream (2.59 × 104 copies g-1 and 1.58 × 103 copies g-1, respectively) and downstream parts (3.26 × 104 copies g-1 and 1.75 × 103 copies g-1, respectively)” to “This study found that the abundance of both comammox clade A and comammox clade B (3.31 × 104 copies g-1 and 2.01 × 103 copies g-1, respectively) in the midstream of the reservoir (M3-M7) were slightly higher than that in the upstream (2.59 × 104 copies g-1 and 1.58 × 103 copies g-1, respectively) and downstream parts (3.26 × 104 copies g-1 and 1.75 × 103 copies g-1, respectively)”
10) Line 429: Is the +/- value correct? Seems like it is off based on value in Line 245.
Answer: Thanks for the reviewer's correction. We apologize for the error here. Changed “The abundance of comammox Nitrospira clade A was the highest (3.00 × 104 ± 32.72 copies/g)” to “The abundance of comammox Nitrospira clade A was the highest (3.00 × 104 ± 8782.37 copies/g)”

Reviewer 2 Report
Authors have improved the text intensively. In the present form the manuscript may be accepted.
Author Response
Thanks to the reviewers for their guidance and help.